# Prevalence and Impact of Cerebral Microbleeds on Clinical and Safety Outcomes in Acute Ischaemic Stroke Patients Receiving Reperfusion Therapy: A Systematic Review and Meta-Analysis

**DOI:** 10.3390/biomedicines11102865

**Published:** 2023-10-23

**Authors:** Shraddha Tipirneni, Peter Stanwell, Robert Weissert, Sonu M. M. Bhaskar

**Affiliations:** 1Global Health Neurology Lab, Sydney, NSW 2150, Australia; 2UNSW Medicine and Health, South Western Sydney Clinical Campuses, University of New South Wales (UNSW), Sydney, NSW 2170, Australia; 3Neurovascular Imaging Laboratory, Ingham Institute for Applied Medical Research, Clinical Sciences Stream, Sydney, NSW 2170, Australia; 4School of Health Sciences, University of Newcastle, Newcastle, NSW 2308, Australia; 5Department of Neurology, Regensburg University Hospital, University of Regensburg, 93053 Regensburg, Germany; 6NSW Brain Clot Bank, NSW Health Pathology, Sydney, NSW 2170, Australia; 7Department of Neurology & Neurophysiology, Liverpool Hospital & South Western Sydney Local Health District (SWSLHD), Liverpool, NSW 2170, Australia

**Keywords:** cerebral microbleeds, stroke, meta-analysis, haemorrhage, reperfusion therapy, prevalence

## Abstract

Background: Cerebral microbleeds (CMBs), a notable neuroimaging finding often associated with cerebral microangiopathy, demonstrate a heightened prevalence in patients diagnosed with acute ischemic stroke (AIS), which is in turn linked to less favourable clinical prognoses. Nevertheless, the exact prevalence of CMBs and their influence on post-reperfusion therapy outcomes remain inadequately elucidated. Materials and Methods: Through systematic searches of PubMed, Embase and Cochrane databases, studies were identified adhering to specific inclusion criteria: (a) AIS patients, (b) age ≥ 18 years, (c) CMBs at baseline, (d) availability of comparative data between CMB-positive and CMB-negative groups, along with relevant post-reperfusion therapy outcomes. The data extracted were analysed using forest plots of odds ratios, and random-effects modelling was applied to investigate the association between CMBs and symptomatic intracerebral haemorrhage (sICH), haemorrhagic transformation (HT), 90-day functional outcomes, and 90-day mortality post-reperfusion therapy. Results: In a total cohort of 9776 AIS patients who underwent reperfusion therapy, 1709 had CMBs, with a pooled prevalence of 19% (ES 0.19; 95% CI: 0.16, 0.23, *p* < 0.001). CMBs significantly increased the odds of sICH (OR 2.57; 95% CI: 1.72; 3.83; *p* < 0.0001), HT (OR 1.53; 95% CI: 1.25; 1.88; *p* < 0.0001), as well as poor functional outcomes at 90 days (OR 1.59; 95% CI: 1.34; 1.89; *p* < 0.0001) and 90-day mortality (OR 1.65; 95% CI: 1.27; 2.16; *p* < 0.0001), relative to those without CMBs, in AIS patients undergoing reperfusion therapy (encompassing intravenous thrombolysis [IVT], endovascular thrombectomy [EVT], either IVT or EVT, and bridging therapy). Variations in the level of association can be observed among different subgroups of reperfusion therapy. Conclusions: This meta-analysis underscores a significant association between CMBs and adverse postprocedural safety outcomes encompassing sICH, HT, poor functional outcome, and increased mortality in AIS patients undergoing reperfusion therapy. The notable prevalence of CMBs in both the overall AIS population and those undergoing reperfusion therapy emphasizes their importance in post-stroke prognostication.

## 1. Introduction

Cerebral microbleeds (CMBs), tiny areas of intracerebral bleeding detected through brain imaging scans such as T2*-weighted Gradient Recalled Echo (T2*GRE) [1] and susceptibility weighted-imaging (SWI) [2], have emerged as pivotal factors linked to adverse outcomes in acute ischemic stroke (AIS), particularly when their prevalence is high [3]. Conventional magnetic resonance imaging (MRI) plays a crucial role in the detection and evaluation of cerebral small vessel disease (CSVD). Key MRI findings indicative of CSVD encompass CMBs, leukoaraiosis [4,5], recent clinically symptomatic subcortical lacunar infarcts, clinically silent lacunes, conspicuous perivascular spaces, and atrophy [6,7]. The increased mortality and poor functional outcomes following stroke can be attributed to the underlying pathophysiological mechanisms involving hypertensive arteriopathy and cerebral amyloid angiopathy (CAA) [8]. Given the substantial burden of CMBs in stroke patients, discerning those presenting with stroke symptoms and concomitant CMBs becomes paramount [9]. This holds exceptional importance in the context of stroke therapies, including intravenous thrombolysis (IVT) and more recently endovascular thrombectomy (EVT) [10] as well as bridging therapies combining IVT and EVT [11]. Tailoring treatments based on clinical considerations and the presence of CMBs has the potential to optimise patient outcomes. Nevertheless, there exists a gap in our understanding of the repercussions of CMBs on reperfusion outcomes post-AIS and their prevalence exclusively in AIS patients. Recognising the significance of CMB prevalence in AIS spans several dimensions, encompassing their role as prognostic indicators for poorer clinical outcomes [12], insights into underlying vascular pathologies [13], the ability to construct predictive models for risk stratification [14] and facilitating well-informed decisions regarding appropriate AIS treatment [15]. Addressing these gaps, our meta-analysis endeavours to elucidate the impact of CMBs on reperfusion outcomes among AIS patients and to quantify the prevalence of CMBs in this specific population.

## 2. Objectives

This study aims to determine the prevalence of CMBs in AIS patients, both in those undergoing reperfusion therapies like IVT, EVT, or bridging therapy. Additionally, it aims to explore the association of CMBs with clinical outcomes among AIS patients undergoing reperfusion therapy. The research will address the following questions:What is the overall prevalence of CMBs among AIS patients?What is the prevalence of CMBs in patients undergoing reperfusion therapy?Do CMBs in AIS patients associate with 90-day functional outcomes?Do CMBs in AIS patients associate with 90-day mortality?Do CMBs in AIS patients associate with risk of symptomatic intracerebral haemorrhage (sICH) or haemorrhagic transformation (HT)?

## 3. Materials and Methods

### 3.1. Literature Search: Study Identification and Selection

Studies were sourced from online databases, including PubMed, Embase, and Cochrane Central Register of Controlled Trials (CENTRAL), spanning the period from 1 January 2000, to June 2023. A comprehensive search strategy was employed, utilising relevant terms such as “cerebral microbleeds”, “microhemorrhages”, “microbleeds”, “microbleed”, and “microhemorrhage”, combined with terms such as “acute ischemic stroke”, “stroke”, “cerebrovascular ischemia”, “brain ischemia”, or “stroke, acute”. The Appendix A provides a detailed overview of the search strategy (Search Strategy). Only research conducted in the English language and involving human participants was considered. Manual screening of reference lists from relevant articles, systematic reviews, and meta-analyses was also conducted to identify further pertinent studies. The Preferred Reporting Items for Systematic Reviews and Meta-analyses (PRISMA) flowchart (Figure 1) illustrates the search process and the inclusion of studies and subgroups, while adhering to reporting standards. In adherence to reporting standards, the PRISMA 2020 (Appendix A) and Meta-analysis of Observational Studies in Epidemiology (MOOSE) checklist is provided in the online Appendix A.

### 3.2. Inclusion and Exclusion Criteria

Studies considered eligible for inclusion met the following criteria: (a) patients diagnosed with AIS; (b) patients aged ≥ 18 years; (c) availability of data on AIS patients exhibiting CMBs; (d) consecutive patients undergoing reperfusion therapy, including IVT, EVT or bridging therapy, and (e) studies with robust methodological design deemed as a minimum sample of 20 patients. Exclusion criteria encompassed: (a) systematic reviews, meta-analyses, and case reports; (b) studies involving animal subjects; (c) studies lacking full-text availability; (d) studies lacking baseline CMB data, (e) studies involving intra-arterial thrombolysis, and (f) studies published in languages other than English or duplicate studies. 

### 3.3. Data Extraction

Using Endnote (Clarivate Analytics, London, UK), we meticulously screened titles and abstracts to identify articles meeting our inclusion criteria. Two researchers conducted the screening process independently, and any disagreements were resolved through consensus discussions. Subsequently, a detailed review of these articles was conducted to assess their eligibility based on predefined criteria. Pertinent data from each selected study were extracted, employing a data extraction sheet on Google Sheets. This sheet encompassed essential details, including: (1) baseline demographics, comprising of author names, country of origin, and year of publication; (2) characteristics of the study population, such as the number of patients with CMBs at baseline, overall cohort size, MRI sequence type, age distribution of patients with and without CMBs, and specific characteristics of patients with AIS; (3) the specific type of reperfusion therapy used; (4) outcome measures encompassing various assessments, such as the modified Rankin Scale (mRS) score for functional outcomes at 90 days, 90-day mortality rate, and the incidence of sICH and HT. The definition of sICH followed established criteria, including the European-Australasian Acute Stroke Study (ECASS), the Second European-Australasian Acute Stroke Study (ECASS-II), the Third European-Australasian Acute Stroke Study (ECASS-III), Safe Implementation of Thrombolysis in Stroke-Monitoring Study (SITS-MOST), National Institute of Neurological Disorders and Stroke (NINDS), and Prolyse in Acute Cerebral Thromboembolism trial two (PROACT-II). The reported definitions of HT and its detection timeframes varied across studies. Through the rigorous methodology, we ensured a comprehensive data collection process that underpinned the robustness of our analysis and the validity of our conclusions.

### 3.4. Methodological Quality Assessment of Included Studies

The methodological quality assessment was performed using the modified Jadad analysis (MJA). The risk funding bias was also noted and was based on author disclosures of funding and any conflicts of interest. These can be accessed in the online Appendix A.

### 3.5. Statistical Analyses

Statistical analyses were performed using STATA v 13.0 (StataCorp, College Station, TX, USA). Baseline characteristics were extracted from the included studies and means and standard deviations (SD) were derived from medians and interquartile ranges (IQR) when needed, following the method outlined by Wan et al. [16]. The pooled prevalence of cerebral microbleeds (CMBs) was estimated using the Metaprop command in STATA, employing a random-effects model. Refined 95% confidence intervals (95% CI) were calculated using the “cimethod (exact)” and “ftt” commands. To assess the association between CMBs in AIS patients and outcomes such as sICH, HT, poor functional outcome, and 90-day mortality, a DerSimonian and Laird (DL) random effects meta-analysis was performed. This analysis was restricted to studies reporting baseline CMB data and outcome related to CMB presence or absence. Summary effects and measures of heterogeneity were tabulated. Forest plots depicting odds ratios (OR) and risk ratios (RR) were generated for association studies (Appendix A) to visually present the data. Heterogeneity among studies was assessed using the I^2^ statistic and *p*-value. The *metaninf* package in STATA was used to evaluate the impact of excluding individual studies on pooled ORs. For potential publication bias, Egger’s test and funnel plots were implemented from the *metabias* and *metafunnel* packages. Asymmetry in the funnel plot was considered, supplemented by the *p*-value from Egger’s test. Subgroup analyses were conducted for IVT, EVT, and bridging therapies, investigating potential treatment specific variations. Cochran’s Q test *p*-values were also considered, and between-study variances were estimated using Tau-squared. A significance threshold of *p* < 0.05 was set for meaningful associations. This rigorous approach ensures the robustness of conclusions drawn and the identification of potential trends in the data.

## 4. Results

### 4.1. Description of Included Studies

In this meta-analysis, we delved into the prevalence of CMBs in AIS patients who underwent reperfusion therapy, involving data from 26 studies comprising 9776 patients. Among these studies, 21 focused on patients undergoing IVT [11,15,17,18,19,20,21,22,23,24,25,26,27,28,29,30,31,32,33,34,35,36], seven on EVT [11,21,22,25,37,38,39,40], 1 on IVT or EVT [21] and one on bridging therapy [11]. Certain studies were excluded from the meta-analysis due to various reasons, such as overlapping patient cohorts, the reporting of CMBs as newly occurring post-reperfusion, and inclusion of patients with Transient Ischemic Attack (TIA). Of the entire AIS patient cohort considered, 6172 individuals exhibited baseline CMBs. Among those undergoing reperfusion therapy, this number was reduced to 1709.

Table 1, Table 2 and Table 3 outline the clinical characteristics and outcomes of the participants across the studies. Moreover, Table 4 provides insights into heterogeneity and summary effects corresponding to specific clinical and safety outcome parameters. An in-depth assessment of the methodological quality and potential funding bias is presented in Appendix A. Additionally, we examined the effect sizes for key outcomes such as functional outcome, mortality at 90 days, sICH and HT, as encapsulated in Table 4. Studies demonstrated minimal potential for publication bias, and this was further reinforced by Egger’s test, as shown in Appendix A. 

### 4.2. Prevalence of CMBs in AIS 

In the context of reperfusion therapy, the overall prevalence of CMBs was found to be 19%. Subgroup analysis based on the type of reperfusion therapy (as shown in Figure 2) revealed that patients undergoing IVT had the highest pooled prevalence at 20% (95% CI: 0.16; 0.25). EVT patients followed with a prevalence of 18% (95% CI: 0.14; 0.23), while bridging therapy exhibited a prevalence of 9% (95% CI: 0.03; 0.18), albeit relying on a limited dataset from a single study, thus warranting cautious interpretation. Notably, significant heterogeneity persisted within these subgroups, resulting in an overall I^2^ of 94.02%. These findings shed light on the variations in CMB prevalence at baseline across different continents and types of reperfusion therapies, providing valuable insights into the epidemiology of CMBs in AIS patients.

### 4.3. Association of CMBs with sICH Post-Reperfusion Therapy

The meta-analysis included a total of 13 studies [11,15,18,19,21,23,25,26,27,28,29,36,40], comprising 5499 patients, aiming to explore the association between the presence of CMBs and sICH. Various criteria were used to define sICH, including ECASS-I, ECASS-II, ECASS-III, NINDS, PROACT-II, and SITS-MOST. The prevalence of sICH using these criteria is shown in Appendix A. In cases where one study utilised multiple definitions, they were considered as separate studies, labelled as ‘a’, ‘b’, ‘c’, and so on. When using the ECASS-III definition, the association of sICH with CMBs in AIS following IVT was the highest, with an OR of 4.12 (95% CI: 1.04; 16.40, *p* < 0.0001). When using the PROACT-II definition, the association of sICH with CMBs in AIS following IVT was the lowest, with an OR of 1.42 (95% CI: 0.21; 9.79, *p* = 0.724), but failed to reach statistical significance. The overall OR for the association between CMB presence and sICH was found to be 2.57 (95% CI: 1.72; 3.83, *p* < 0.0001). Further stratification was performed based on the type of reperfusion therapy as shown in Figure 3. Among these, CMBs in IVT patients demonstrated the strongest association with sICH, with an OR of 2.57 (95% CI: 1.82; 3.61, *p* = 0.045). On the other hand, EVT showed increased odds of sICH, but this result was not statistically significant, with an OR of 1.14 (95% CI: 0.40; 3.21, *p* = 0.805). For CMBs in those who underwent bridging therapy, the association with sICH was not statistically significant, with an OR of 0.67 (95% CI: 0.03; 13.28, *p* = 0.792). For a study with either EVT or IVT, CMBs were highly associated with sICH, with an OR of 8.96 (95% CI: 4.82; 16.63, *p* < 0.0001) [21]. It is crucial to note that only one study on bridging therapy and CMBs was available, which limits the certainty of this finding. The overall heterogeneity in the meta-analysis was low, with an I^2^ of 0.0%, however, this result was not statistically significant (*p* = 0.079). Minimal publication bias was observed from the inspection of the funnel plot (Appendix A) and confirmed by Egger’s test (Appendix A). These findings support a significant association between CMB presence and sICH, particularly with IVT, while the results for EVT and bridging therapy were not statistically significant due to limited data availability and lower sample sizes. The low heterogeneity and minimal publication bias enhance the reliability and validity of the results.

### 4.4. Association of CMBs with HT Post-Reperfusion Therapy

The meta-analysis included a total of 10 studies [19,20,21,24,26,27,28,29,31,38,39], encompassing 3882 patients. The presence of CMBs was associated with higher odds of HT for those undergoing reperfusion therapy as shown in Figure 3 (OR 1.53; 95% CI: 1.25; 1.88; *p* < 0.0001). Similarly, within the subgroups of patients who received IVT or EVT, there were increased odds of HT, but only IVT and IVT/EVT reached statistical significance (IVT: OR 1.46; 95% CI: 1.03; 2.07; *p* = 0.034; IVT/EVT: OR 1.92; 95% CI: 1.37; 2.68; *p* < 0.0001; EVT: OR 1.19; 95% CI: 0.81; 1.76; *p* = 0.373). The overall heterogeneity in the meta-analysis was low, with an I^2^ of 0.0%, and this finding was not statistically significant (*p* = 0.768). Moreover, minimal publication bias was observed (Appendix A). While the presence of CMBs showed a trend towards higher odds of HT, the association did not reach statistical significance patients receiving EVT only. The low heterogeneity and minimal publication bias enhance the reliability and validity of these findings. However, further research with larger sample sizes may be needed to establish the significance of the observed trends.

### 4.5. Association of CMBs with 90-Day Functional Outcomes Post-Reperfusion

The meta-analysis included a total of seven studies [11,15,18,19,21,25,38], involving 4199 participants. Poor functional outcome at 90 days was uniformly defined as an mRS score of 3–6 across all studies. Overall, the presence of CMBs was associated with significantly increased odds of poor functional outcome at 90 days for those undergoing reperfusion therapy (OR 1.59; 95% CI: 1.34; 1.89; *p* < 0.0001). However, upon further examination of the subgroups as shown in Figure 4, bridging therapy was associated with increased odds of poor functional outcome (OR 2.77; 95% CI: 0.47; 16.27; *p* = 0.260) although this association did not reach statistical significance. It is crucial to note that this finding was based on data from only one study, which may limit its validity. In contrast, both IVT and EVT demonstrated statistically significant associations with poor functional outcome at 90 days. For IVT, the odds ratio was 1.70 (95% CI: 1.28; 2.25; *p* < 0.0001), while for EVT, it was 1.70 (95% CI: 1.26; 2.29; *p* = 0.001). For IVT/EVT, it was 1.32 (95% CI: 0.95; 1.84; *p* = 0.103), the association was not statistically significant. The overall heterogeneity in the meta-analysis was low, with an I^2^ of 0.0%, and this result was not statistically significant (*p* = 0.626). Furthermore, minimal publication bias was observed (Appendix A). More research is needed, especially with larger sample sizes, to further elucidate the significance of these associations.

### 4.6. Association of CMBs with 90-Day Mortality Post-Reperfusion Therapy

Overall, five studies [11,15,18,21,25] were included in the meta-analysis mortality at 90 days, consisting of 3330 patients. There was a strong association between CMBs and mortality at 90 days for those undergoing reperfusion therapy (OR 1.65; 95% CI: 1.27; 2.16; *p* < 0.0001). Within the subgroups as shown in Figure 4, bridging therapy showed no association with mortality at 90 days (OR 0.67; 95% CI: 0.07; 6.37; *p* = 0.740) although this association did not reach statistical significance. IVT/EVT showed an association with mortality at 90 days (OR 1.40; 95% CI: 0.89; 2.20; *p* = 0.142) but also failed to reach statistical significance. It is crucial to note that both findings were based on data from only one study, which may limit their validity. Similarly, for IVT, the OR was 1.52 (95% CI: 0.91; 2.54; *p* = 0.104) meaning the association did not reach statistical significance. In contrast, EVT demonstrated a statistically significant association with mortality at 90 days. OR: 2.14 (95% CI: 1.37; 3.34; *p* = 0.001). The overall heterogeneity in the meta-analysis was low, with an I^2^ of 0.0%, and this result was not statistically significant (*p* = 0.721). Some publication bias was observed in the meta-analysis (Appendix A). However, it is essential to acknowledge that further research is warranted to gain a deeper understanding of the significance of these associations.

## 5. Discussion

The findings of this meta-analysis provide valuable insights into the association between CMBs and post-thrombolysis outcomes in AIS patients as well as prevalence based on several studies. The results indicate that CMBs are significantly associated with increased odds of sICH, HT, and poor functional outcomes and death at 90 days post-reperfusion. Notably, the significance of the association between CMBs and HT appeared to differ between IVT and EVT subgroups, with slightly increased odds of HT after IVT compared to EVT. In summary, this meta-analysis highlights the importance of personalised and evidence-based treatment decisions in AIS patients. It underscores the significance of considering individual patient characteristics, such as the presence of CMBs, when selecting the most appropriate treatment strategy. By identifying patients at higher risk of adverse outcomes, clinicians can optimise therapeutic approaches, improve patient outcomes and minimise complications.

Pathophysiological reasons for the observed outcomes could be multifactorial. The presence of CMBs may indicate underlying vascular pathology such as CAA [13], hypertensive arteriopathy [13] or other cerebral small vessel disease [41], which could predispose patients to a higher risk of haemorrhagic complications after thrombolysis [42]. Blood brain barrier (BBB) dysfunction can occur with CMB presence due to inflammation and microvascular injury [43]. Damage to the BBB causes hyperpermeability which allows the entry of inflammatory cells, cytokines which can damage brain parenchyma [43]. After acute ischaemic stroke, BBB dysfunction can increase the risk of HT due to inflammatory molecules and blood leaking into the brain tissue [44]. This can worsen the functional outcome after stroke by concurrently increasing oedema, causing neurological deficits and mass effect [45]. Reperfusion therapy can activate inflammatory responses [46] and due to pre-existing inflammation in those who have CMBs, this could lead to greater parenchymal damage and poorer outcomes [21]. As CMBs are suggestive of underlying small vessel disease, the compromised vessels after stroke may be at greater risk of rupture after reperfusion therapy [47], leading to sICH [48] or HT [49]. Moreover, CMBs might serve as a marker for more severe or advanced cerebrovascular disease [50], which could contribute to poorer functional outcomes [51] and higher mortality rates [52]. Understanding the underlying pathophysiology of CMBs and their implications in the context of reperfusion therapies is a vital area for future research.

The pooled prevalence of CMBs in AIS patients receiving reperfusion therapy overall was 19%. This is within the range of other studies such as the Egyptian elderly with CMBs having a prevalence of 29.4% [53], Chinese population with 24% [54] and from other studies that were used in the prevalence meta-analysis [11,15,17,18,19,20,21,22,23,24,25,26,27,28,29,30,31,32,33,34,35,36,37,38,39,40,54,55,56,57,58,59,60,61,62,63,64,65,66,67,68,69,70,71,72,73,74,75,76,77,78,79,80,81,82,83,84]. A previous meta-analysis had a crude prevalence of 23.4% and a prevalence of 18.9% for 1–10 CMBs and 0.8% for >10 CMBs [85]. This is slightly higher than our analysis however, our meta-analysis had looked at CMBs in patients specifically undergoing reperfusion therapy. When stratified by continent, the prevalence was the highest in Asian countries, with the majority of the data coming from East Asia. Genetics are unlikely to explain the ethnic difference as other factors such as age and comorbidities would contribute to heterogeneity. Data on CMB prevalence comparing ethnicities is limited. In less developed countries, it is difficult to estimate CMB prevalence in AIS due to the paucity of studies and likely less access to diagnostic equipment for CMBs such as SWI or T2*GRE MRI [86]. 

We observed variations in impacts of different reperfusion therapies on the relationship between CMBs and outcomes. IVT was significantly associated with sICH, HT, poor functional outcomes, and mortality. Conversely, there were statistically significant association between CMBs and outcomes in EVT, specifically in terms of poor functional outcomes and 90-day mortality. However, no such significant associations were found for sICH and HT. The treatment of AIS patients with CMBs with bridging therapy was not associated with clinical and safety outcomes, possibly due to the limited availability of data for these subgroups. It is essential to recognise the limitation of having only one study providing data on CMBs and bridging therapy outcomes, which may affect the accuracy and generalisability of this specific finding.

Our study confirms that CMBs are associated with worse clinical outcomes following reperfusion therapy, consistent with findings reported in previous meta-analyses [85,87]. Our findings show that there is a negative association between CMBs in AIS patients who underwent reperfusion therapy and worse clinical outcomes such as 90-day poor functional outcome (mRS 3–6) and 90-day mortality. However, more data on EVT and bridging therapy is needed as most of the studies included IVT patients. 

In terms of adverse outcomes, our study suggests that AIS patients with CMBs are associated with increased odds of sICH and HT. This aligns with previous studies, in which the presence of CMBs were associated with sICH and HT [49,87]. Prior CMBs are generally associated with microangiopathy in the brain, making it prone to haemorrhage and thus sICH or HT [88]. One of the studies included in our meta-analysis found that CMB presence alone was not significantly associated with an increased risk of HT or sICH [28]. In five of the 10 studies included in the meta-analysis, CMB presence was not associated with the increased risk of HT [20,27,28,39,40]. However, studies such as Charidimou et al. (2015) [89], Tsivgoulis et al. (2016) [85] and Charidimou et al. (2017) [12] found that CMB presence was associated with HT. This could be explained due to the low numbers of patients with high CMB burden included in the studies for the meta-analysis, as high CMB burden is associated with adverse outcomes such as sICH [19]. In Yan et al. (2015), CMB presence was not associated with increased risk of HT after IVT but was associated with parenchymal haemorrhage (PH) and poor functional outcomes for ≥3 CMBs [35]. As there are discrepancies in the data, further studies on reperfusion therapy in patients with high CMB burdens are needed to assess post-stroke outcomes in order to guide decision making. 

The future implications of these findings lie in the potential optimisation of thrombolysis strategies in AIS patients with CMBs. The significant association between CMBs and adverse outcomes, particularly following IVT, underscores the importance of careful consideration and risk stratification when determining the appropriateness of thrombolysis as a treatment option. Neurologists may need to carefully weigh the potential benefits of reperfusion therapy against the increased risks of sICH and unfavorable functional outcomes in patients with CMBs. However, CMBS should not serve as a barrier to treatment. Additionally, further research with larger sample sizes is crucial to solidify the significance of the associations between CMBs and outcomes for EVT and bridging therapy. While this meta-analysis provides valuable insights, further research is needed to establish the significance of these observed associations, particularly for EVT and bridging therapy. Further investigations into the underlying pathophysiological mechanisms and larger-scale studies can offer a more comprehensive understanding of the relationship between CMBs and post-thrombolysis outcomes, guiding clinicians towards more effective and personalised treatment strategies for AIS patients. Given the distinct demographics and risk factors in very old age segment (aged 80 years or older) of stroke patients, future clinical studies are warranted to investigate the prevalence and effects of CMBs on clinical and safety outcomes in the specific subgroup of AIS patients [90]. Resuming anticoagulation, and its optimal timing, after an ICH in patients with atrial fibrillation (AF) poses an ongoing clinical conundrum [91]. Individualized treatment decision-making that factors in the potential for future bleeding events and the risk of thromboembolic complications is suggested that takes into account various risk factors, including blood pressure control, age, the location of the ICH, the presence of CSVD markers (such as CMBs, leukoaraiosis, and cortical superficial siderosis), and the indication for antiplatelet therapy [92].

## 6. Limitations

This study has several limitations that warrant consideration. Firstly, the inclusion of the studies involving patients receiving IVT in conjunction with other therapies such as antithrombotic therapies. This could potentially impact outcomes after AIS. Notably, the varying definitions of sICH across studies, employing criteria such as ECASS-I, ECASS-II, ECASS-III, NINDS, PROACT-II, or SITS-MOST, introduces substantial variability. Different MRI imaging sequences also contributes to variability. SWI outperforms T2*GRE in the detection of CMBs, especially when higher magnetic field strengths are employed [93,94,95]. Variability in results may partly stem from variations in SWI usage and the utilization of higher magnetic field strengths, which enhance CMB detection capabilities. Furthermore, there remains a scarcity of studies specifically reporting post-thrombolytic outcomes for patients undergoing only EVT or bridging therapy. More specifically, bridging therapy outcomes *vis a vis* the prevalence of CMBs and their implications have scarcely been reported, including outcomes such as sICH, mortality at 90 days, functional status assessed via mRS at 90 days, HT, and recurrent stroke incidence. Definitions of HT were not always specified, and follow-up imaging times were variable, contributing to uncertainties in results. Ambiguities emerged in some studies regarding the precise number of participants who underwent IVT, EVT, or both. Additionally, inconsistencies in reporting CMB location and burden parameters were evident, where certain studies combined infratentorial and deep CMBs, while others grouped CMBs under broader classifications such as “1–10 CMBs” rather than distinct categories (1, 2–4, or ≥5). The precise location of CMBs was largely underreported. Gender disparities may exist in the distribution of risk factors, stroke subtype, stroke severity, and post-stroke outcomes [96]. Nonetheless, the present study did not explore this and it merits further investigation. Furthermore, substantial heterogeneity persisted across study cohorts in terms of sample sizes, clinical risk factors, and ethnic backgrounds. These limitations should be considered when interpreting the study results and forming conclusive insights.

## 7. Conclusions

In conclusion, CMBs play a crucial role as prognostic indicators in patients with AIS undergoing reperfusion therapy. This meta-analysis reveals a strong association between CMBs and adverse clinical outcomes, encompassing poor functional outcomes at 90 days, increased 90-day mortality, and an elevated risk of sICH and HT. These consistent findings traverse diverse reperfusion therapies, except for bridging therapy, which lacks substantial data, limited to a single study. The evidence robustly underscores the integration of CMB assessment in clinical decision-making for AIS patients undergoing reperfusion therapy. It is, however, vital to emphasize that the presence of CMBs should not deter the use of thrombolytic therapies. This meta-analysis accentuates the significance of identifying CMBs as pivotal indicators for patient prognosis in the context of reperfusion therapy. Clinicians and healthcare providers may take cognisance of these findings when formulating clinical guidelines to optimise treatment strategies for individuals with AIS. Considering the high prevalence of CMBs in AIS patients receiving reperfusion therapy, it is important to acknowledge CMBs as a prognostic indicator in AIS patients.

## Figures and Tables

**Figure 1 biomedicines-11-02865-f001:**
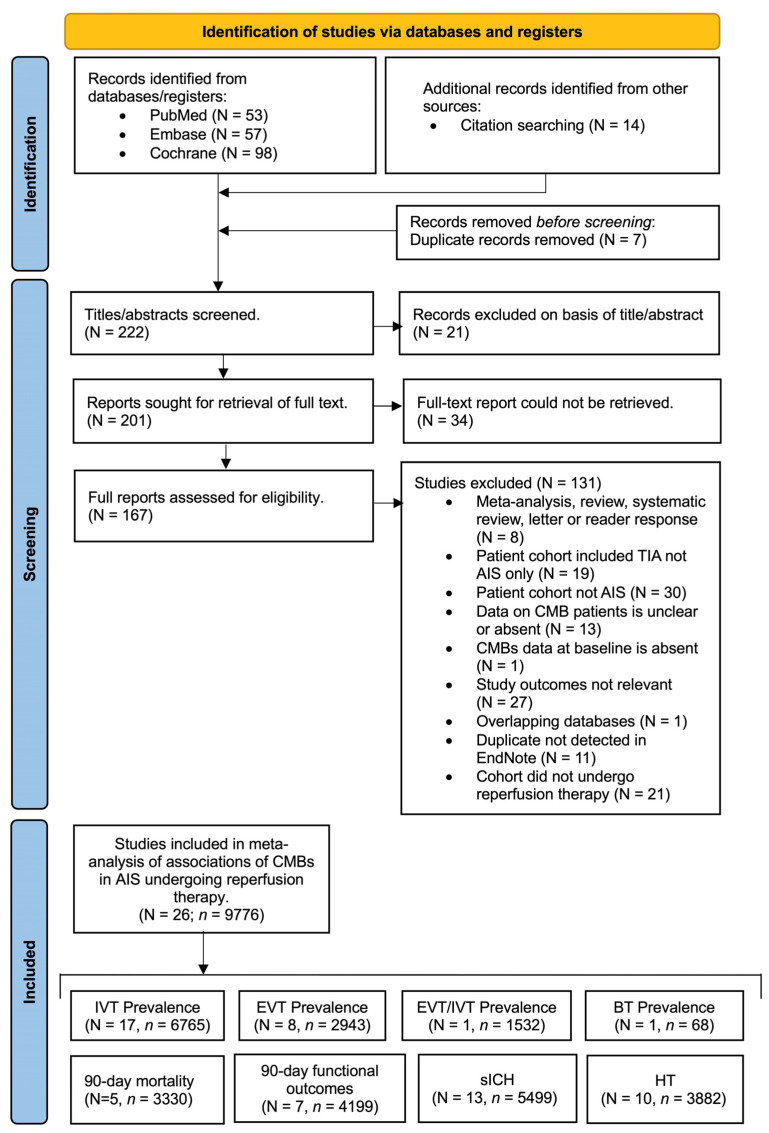
The Preferred Reporting System for Systematic Reviews and Meta-Analyses (PRISMA) flowchart showing the studies included in this meta-analysis. Abbreviations: *n* = number of studies, *n* = total number of patients, CMBs = cerebral microbleeds, IVT = intravenous thrombolysis EVT = endovascular thrombectomy, BT = bridging therapy, sICH = symptomatic intracerebral haemorrhage, HT = haemorrhagic transformation.

**Figure 2 biomedicines-11-02865-f002:**
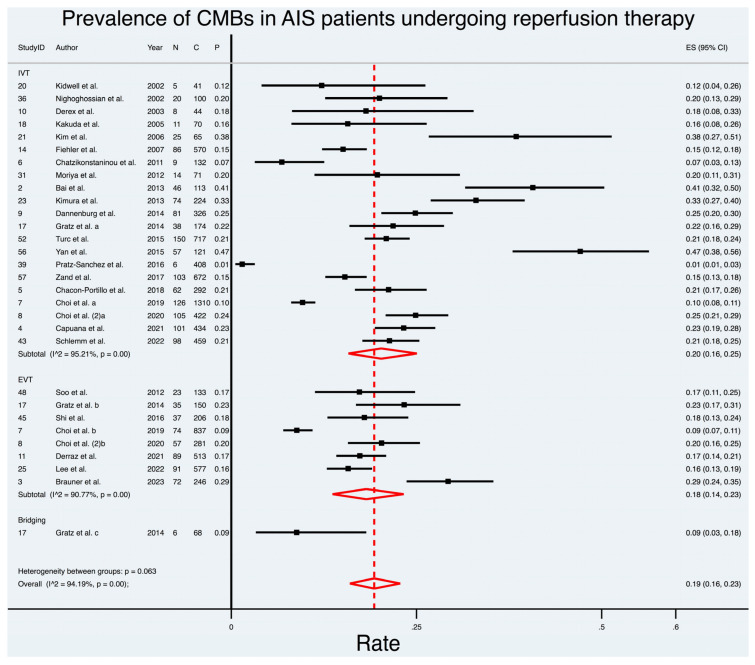
Prevalence of cerebral microbleeds in acute ischemic stroke patients undergoing reperfusion therapy [11,15,17,18,19,20,21,22,23,24,25,26,27,28,29,30,31,32,33,34,35,36,37,38,39,40]. Note: a, b and c correspond to cohorts receiving IVT, EVT and bridging, respectively, in the said study. Choi et al. [21] and Choi et al. (2) [22] refer to studies from 2019 and 2020, respectively. Abbreviations: CMB = cerebral microbleeds, AIS = acute ischaemic stroke, *n* = number of patients with CMBs, C = total cohort number, *p* = prevalence, IVT = intravenous thrombolysis, EVT = endovascular thrombectomy, ES = effect size, I^2^ = heterogeneity value, *p* = *p*-value.

**Figure 3 biomedicines-11-02865-f003:**
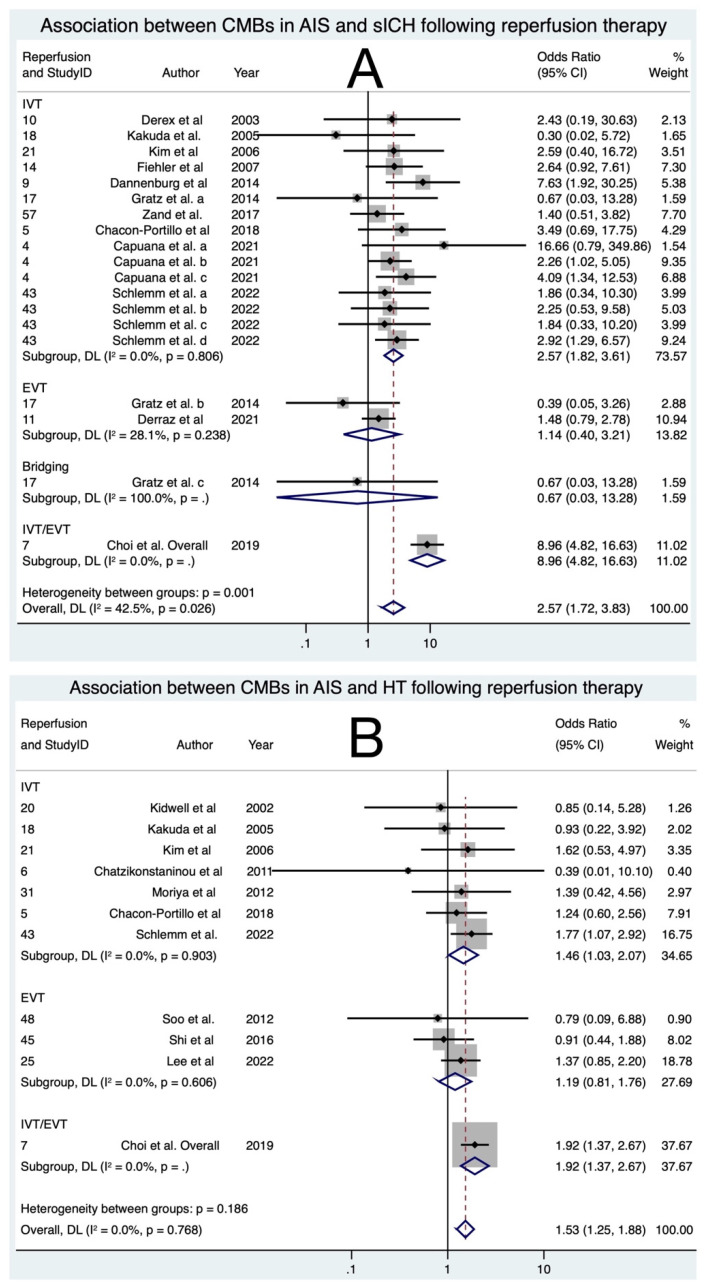
Forest plots of association of cerebral microbleeds with clinical outcomes: (**A**) symptomatic intracerebral haemorrhage (sICH) [11,15,18,19,21,23,25,26,27,28,29,36,40] and (**B**) haemorrhagic transformation (HT) [19,20,21,24,26,27,28,29,31,38,39] in acute ischemic stroke patients undergoing reperfusion therapy. Note: Capuana et al. a, b, c corresponds to cohorts assessed for sICH using the SITS-MOST, ECASS-II and NINDS criteria, respectively, in the said study. Schlemm et al. a, b, c, d correspond to cohorts assessed for sICH using the SITS-MOST, ECASS-II, ECASS-III, and NINDS, respectively. Gratz et al. [11] a, b, c and overall represent cohorts receiving IVT, EVT, Bridging and IVT and/or EVT, respectively, within this study. Abbreviations: CMBs = cerebral microbleeds, sICH = symptomatic intracerebral haemorrhage, HT = haemorrhagic transformation, IVT = intravenous thrombolysis, EVT = endovascular thrombectomy, OR = odds ratio, CI = confidence interval, *p* = *p*-value, DL = DerSimmonian and Laird, I^2^ = heterogeneity, ECASS = European Cooperative Acute Stroke Study, ECASS-II = second European Cooperative Acute Stroke Study, ECASS-III = third European Cooperative Acute Stroke Study, NINDS = National Institute of Neurological Disorders and Stroke, SITS-MOST = Safe Implementation of Thrombolysis in Stroke-Monitoring Study, PROACT-II = Prolyse in Acute Cerebral Thromboembolism trial 2.

**Figure 4 biomedicines-11-02865-f004:**
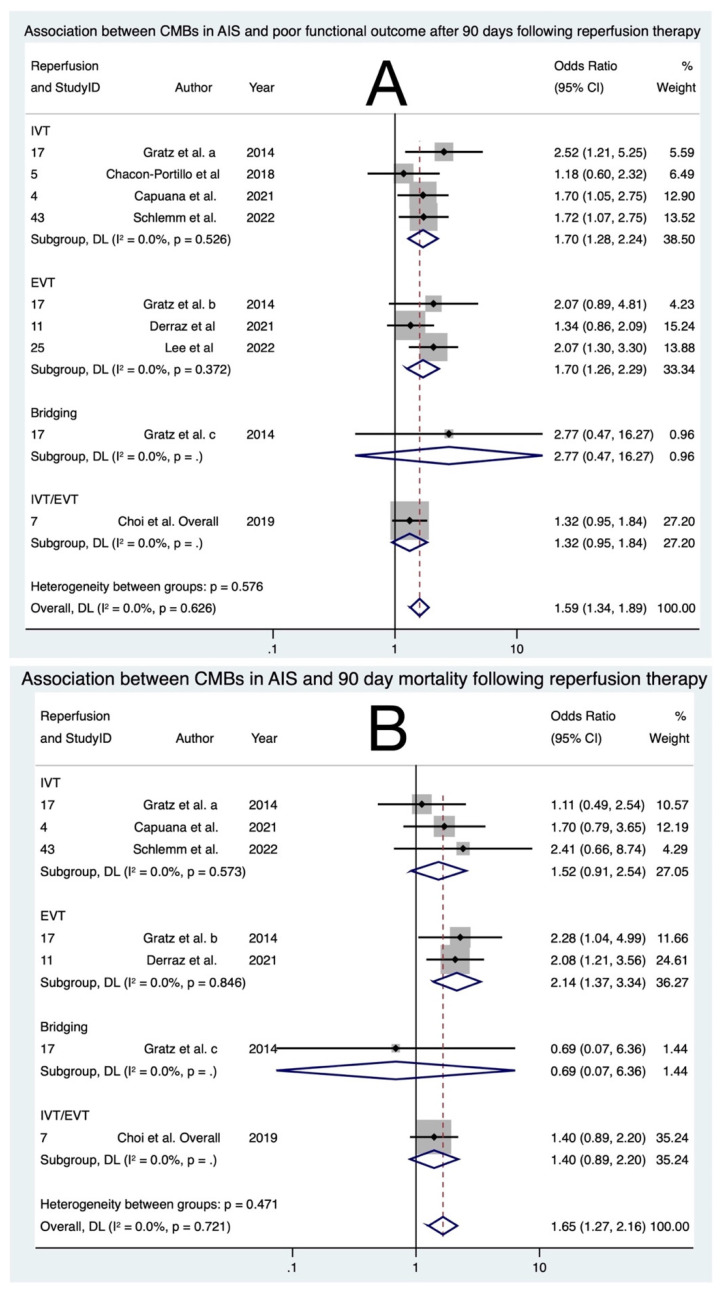
Forest plots of association of cerebral microbleeds with adverse outcomes: (**A**) poor functional outcome at 90 days [11,15,18,19,21,25,38] and (**B**) mortality at 90 days [11,15,18,21,25] in acute ischemic stroke patients undergoing reperfusion therapy. Note: Gratz et al. a, b and c correspond to cohorts receiving IVT, EVT and bridging therapy, respectively, in the said study. Abbreviations: IVT = intravenous thrombolysis, EVT = endovascular thrombectomy, OR = odds ratio, CI = confidence interval, *p* = *p*-value, DL = DerSimmonian and Laird, I^2^ = heterogeneity.

**Table 1 biomedicines-11-02865-t001:** Clinical characteristics of studies selected for meta-analysis for acute ischemic stroke patients who underwent reperfusion therapy.

Author	Year	Continent	Study Type	Number of CMBs	Cohort	Male % Overall			Age Mean (SD)			Baseline NIHSS			Reperfusion	CMB Imaging	MRI Follow-Up Time	HT Detection Timeframe	HT Imaging Modality	sICH Definition	sICH Detection Timeframe	sICH Imaging Modality
						Overall	CMB	No CMB	Overall	CMB	No CMB	Overall	CMB	No CMB								
Bai et al. [17]	2013	Asia	Prospective	46	113	NR	NR	NR	NR	NR	NR	NR	2.3	2	IVT	SWI	24 h	24 h	CT	NR	NR	MRI
Brauner et al. [37]	2023	Europe	Prospective	72	246	47.6	NR	NE	73.6 (13.3)	NR	NR	NR	NR	NR	EVT	T2*GRE, SWI	24–36 h	NR	NR	NR	NR	CT
Capuana et al. [18]	2021	Europe	Prospective	101	434	60.8	NR	75	68.3 (13.5)	69.0 (12.6)	68.1 (13.8)	16.3 (6.0)	NR	NR	IVT	T1WI, T2WI, T2*GRE, FLAIR, DWI, PWI, MRA	24 h	NR	NR	SITS-MOST, ECASS-II, NINDS	22–36 h	NR
Chacon-Portillo et al. [19]	2018	North America	Retrospective	62	292	82.2	NR	NR	63 (15)	NR	NR	9.3 (7.4)	9.7 (7.5)	9 (6.7)	IVT	SWI	72 h—6 days	24 h	GRE, NC CT	NINDS	NR	NR
Chatzikonstaninou et al. [20]	2011	Europe	Prospective	9	132	50.8	10.4	90.0	74 (9.9)	74.2 (9.9)	72.7 (7.1)	7.9 (5.5)	NR	NR	IVT	T2*GRE	72 h	NR	NR	NR	NR	NR
Choi et al. a [21]	2019	Asia	Prospective	126	1310	NR	NR	NR	NR	NR	NR	NR	NR	NR	IVT	T2*GRE	7 days	24 h	GRE, CT	ECASS-I	Within 7 days	NR
Choi et al. b [21]	2019	Asia	Prospective	74	837	NR	NR	NR	NR	NR	NR	NR	NR		EVT	T2*GRE	7 days	24 h	GRE, CT	ECASS-I	Within 7 days	NR
Choi et al. overall [21]	2019	Asia	Prospective	165	1532	55.8	9.2	90.8	69.2 (NR)	72 (11.2)	68.9 (11.9)	6.7 (7.4)	12.3 (4.8)	11 (5.1)	IVT and/or EVT	T2*GRE	7 days	24 h	GRE, CT	ECASS-I	Within 7 days	NR
Choi et al. (2)a [22]	2020	Asia	Prospective	105	422	NR	NR	NR	NR	NR	NR	NR	NR	NR	IVT	T2*GRE	NR	NR	NR	NR	NR	NR
Choi et al. (2)b [22]	2020	Asia	Prospective	57	281	NR	NR	NR	NR	NR	NR	NR	NR	NR	EVT	T2*GRE	NR	NR	NR	NR	NR	NR
Choi et al. (2) Overall [22]	2020	Asia	Prospective	393	1742	53.1	20.6	79.4	72.6 (NR)	73.2 (NR)	72.5 (NR)	NR	8 (7.4)	6.7 (7.4)	IVT and/or EVT	T2*GRE	NR	NR	NR	NR	NR	NR
Dannenburg et al. [23]	2014	Europe	Prospective	81	326	NR	NR	NR	76 (11.9)	NR	NR	9 (6.7)	NR	NR	IVT	T2*GRE	36 h	NR	NR	ECASS-III	NR	T2WI
Derex et al. [24]	2004	Europe	Prospective	8	44	NR	NR	NR	63.2 (14.1)	NR	NR	14 (5.8)	NR	NR	IVT	T2*GRE	24 h and 7th day	NR	NR	PROACT-II^⍬^	NR	CT
Derraz et al. [25]	2021	Europe	Prospective	89	513	47.4	16.5	83.5	69.4 (25.9)	80.8 (15.7)	67.3 (25.4)	16 (7.4)	17.7 (6.8)	15.7 (8.2)	EVT	T2*GRE	NR	NR	NR	ECASS-II	24 h	NR
Fiehler et al. [26]	2007	Multi	Retrospective	86	570	59.8	NR	NR	68.3 (13.3)	NR	NR	NR	NR	NR	IVT	T2*GRE	6 h	Clinical deterioration—10 days	NR	ECASS-I	1–10 days	T2WI
Gratz et al. a [11] *	2014	Europe	Prospective	38	174	NR	NR	NR	NR	NR	NR	8 (4.5)	NR	NR	IVT	T2*GRE	72 h	72 h	MRI, NCCT with CTA	PROACT-II ^⍬^	72 h	MRI, NCCT with CTA
Gratz et al. b [11] *	2014	Europe	Prospective	35	150	NR	NR	NR	NR	NR	NR	12.7 (6.7)	NR	NR	EVT	SWI	72 h	72 h	MRI, NCCT with CTA	PROACT-II ^⍬^	72 h	MRI, NCCT with CTA
Gratz et al. c [11] *	2014	Europe	Prospective	6	68	NR	NR	NR	NR	NR	NR	NR	NR	NR	Bridging	SWI	72 h	72 h	MRI, NCCT with CTA	PROACT-II ^⍬^	72 h	MRI, NCCT with CTA
Gratz et al. overall [11] *	2014	Europe	Prospective	6	68	56.9	NR	NR	1 (13.7)	NR	NR	15.3 (26.0)	NR	NR	IVT and/or EVT	SWI	72 h	72 h	MRI, NCCT with CTA	PROACT-II ^⍬^	72 h	MRI, NCCT with CTA
Kakuda et al. [27] ^⍦^	2005	Multi	Prospective	11	70	44.3	19.4	NR	NR	70 (32)	71 (29)		NR	NR	IVT	T2*GRE, DWI, MRA, T1WI, FLAIR	30th day	NR	CT, MRI	ECASS-II	36 h	CT, MRI
Kidwell et al. [28]	2002	North America	Prospective	5	41	92.0	NR	NR	NR	NR	NR	NR	NR	NR	IVT	T2WI, T2*GRE, EPI-SWI	NR	Immediately after tPA and 24 h	CT	NR	NR	NR
Kim et al. [29]	2006	Asia	Retrospective	25	65	56.9	NR	NR	NR	67 (NR)	NR	NR	NR	NR	IVT	T2*GRE, DWI, FLAIR, MRA	24–72 h	NR	T2WI	NR	48 h	T2WI
Kimura et al. [30]	2013	Asia	Prospective	74	224	54.0	NR	NR	76.2 (10.6)	NR	NR	NR	11 (9.2)	13 (9.8)	IVT	T1WI, T2WI, T2*GRE, DWI, FLAIR, MRA	24 h	NR	T2WI	SITS-MOST	24 h	T2WI, CT
Lee et al. [38]	2022	Asia	Prospective	91	577	55.8	14.9	85.1	67 (13)	NR	NR	NR	NR	NR	EVT	DWI, T2*GRE	NR	5–7 days	CT	NR	NR	NR
Moriya et al. [31]	2012	Asia	Retrospective	14	71	70.4	NR	NR	NR	NR	NR	NR	NR	NR	IVT	T2*GRE, T1WI, T2WI, DWI, MRA	NR	24 h, 4–7 days	CT	NR	NR	NR
Nighoghossian et al. [32]	2002	Europe	Prospective	20	100	58.0	NR	NR	60 (13)	71.2 (8.6)	56.7 (13)	NR	NR	NR	IVT	DWI, T2*GRE, T2WI, MRA	10 ± 7 h	7th day	NR	NR	NR	NR
Pratz-Sanchez et al. [33]	2016	Europe	Prospective	6	408	52.2	NR	NR	NR	NR	NR	5.2 (5.5)	5.9 (6.1)	4.8 (5.1)	IVT	NR	1–13 days	36 h	CT	NR	NR	NR
Schlemm et al. [15] ^⍧^	2022	Europe	Prospective	98	459	62.8	19.4	80.6	NR	71.7 (NR)	67 (NR)	NR	NR	NR	IVT	SWI, T2*GRE	22–36 h	22–36 h	NR	SITS-MOST, ECASS-II, ECASS-III, NINDS	22–36 h	NR
Shi et al. [39]	2016	Asia	Prospective	37	206	42.2	14.9	85.1	NR	77 (14)	66.7 (9.8)	NR	NR	NR	EVT	T2*GRE, FLAIR	NR	18–36 h	CT, GRE	NR	NR	NR
Soo et al. [40]	2012	Asia	Prospective	23	133	NR	NR	NR	NR	67 (17)	67.4 (14)	NR	NR	16 (5.9)	EVT	T2*GRE, T1WI, T2WI	NR	NR	NR	NR	NR	CT
Turc et al. [34]	2015	Europe	Prospective	150	717	49.0	NR	NR	74 (NR)	NR	NR	NR	NR	NR	IVT	T2*GRE	NR	NR	NR	NINDS, ECASS-II, ECASS-III, SITS-MOST	24 h	NR
Yan et al. [35] ^⍭^	2015	Asia	Prospective	57	121	NR	NR	NR	NR	NR	NR	NR	NR	NR	IVT	DWI, T2*GRE, FLAIR	24 h	NR	GRE	ECASS-II	24 h	GRE
Zand et al. [36]	2017	Multi	Prospective	103	672	51.9	NR	NR	62 (14)	NR	NR	NR	NR	NR	IVT	T2*GRE	24 h	NR	CT	ECASS-II	24 h	MRI

Note: ^⍦^ Kakuda et al. [27]: symptomatic and asymptomatic haemorrhages were categorised based on CT or MRI. Data on sICH or HT imaging was not specified. * Gratz et al. [11]: follow up imaging was performed within 72 h, it did not specify whether that was for sICH or HT. ^⍬^ PROACT-II in Derex et al. [24] Gratz et al. [11] had bleeding complications defined by the PROACT-II trial. ^⍧^ Schlemm et al. [15]: follow up imaging was performed from 22–36 h after treatment, it did not specify whether that was for sICH or HT. ^⍭^ Yan et al. [35]: follow up imaging was performed from 24 h after treatment, it did not specify whether that was for sICH or HT. Gratz [11]/Choi [22] et al. a, b and c correspond to cohorts receiving IVT, EVT and bridging, respectively, in the said study. Choi et al. [21] and Choi et al. (2) [22] refer to studies from 2019 and 2020, respectively. Abbreviations: CMB = cerebral microbleed, Multi = multiple continents, IVT = intravenous thrombolysis, EVT = endovascular thrombectomy, NR = not reported, T2*GRE = T2 Gradient Echo Imaging, T2WI = T2 Weighted Imaging, T1WI = T1 Weighted Imaging, DWI = Diffusion Weighted Imaging, SWI = Susceptibility Weighted Imaging, FLAIR = Fluid Attenuated Inversion Recovery, MRA = Magnetic Resonance Angiography, EPI-SWI = Echo Planar Imaging Susceptibility Weighted Imaging, PWI = Perfusion Weighted Imaging, CT = Computed Tomography, NCCT = Non-Contrast Computed Tomography, CTA = CT Angiography, GRE = Gradient Echo Sequences, NIHSS = National Institutes of Health Stroke Scale, ECASS = European Cooperative Acute Stroke Study, ECASS-II = second European Cooperative Acute Stroke Study, ECASS-III = third European Cooperative Acute Stroke Study, NINDS = National Institute of Neurological Disorders and Stroke, SITS-MOST = Safe Implementation of Thrombolysis in Stroke-Monitoring Study, PROACT-II = Prolyse in Acute Cerebral Thromboembolism trial 2.

**Table 2 biomedicines-11-02865-t002:** Rates of comorbidities cohorts of acute ischemic stroke patients with cerebral microbleeds included in the meta-analysis.

Author	Year	Clinical Risk Factors, *n* (%)						
		AF	HL	HTN	CAD	PS/TIA	Smoking	DM
Bai et al. [17]	2013	NR	NR	NR	NR	NR	NR	NR
Brauner et al. [37]	2023	NR	NR	NR	NR	NR	NR	NR
Capuana et al. [18]	2021	17 (17%)	30 (30%)	77 (76%)	NR	10 (10%) ^a^	34 (34%) ^b^	15 (15%)
Chacon-Portillo et al. [19]	2018	NR	NR	NR	NR	NR	NR	NR
Chatzikonstaninou et al. [20]	2011	NR	NR	NR	NR	NR	NR	NR
Choi et al. a [21]	2019	NR	NR	NR	NR	NR	NR	NR
Choi et al. b [21]	2019	NR	NR	NR	NR	NR	NR	NR
Choi et al. overall [21]	2019	94 (57%)	13 (8%)	123 (75%)	NR	40 (24%)	44 (27%) ^d^	56 (34%)
Choi et al. (2)a [22]	2020	NR	NR	NR	NR	NR	NR	NR
Choi et al. (2)b [22]	2020	NR	NR	NR	NR	NR	NR	NR
Choi et al. (2) overall [22]	2020	393 (100%)	50 (13%),	245 (62%)	NR	106 (27%)	65 (17%) ^c^	85 (22%)
Dannenburg et al. [23]	2014	NR	NR	NR	NR	NR	NR	NR
Derex et al. [24]	2004	NR	2 (25%)	4 (50%)	NR	NR	NR	1 (13%)
Derraz et al. [25]	2021	38 (43%)	36 (40%)	65 (73%)	21 (24%)	19 (21%)	28 (31%) ^c^	14 (16%)
Fiehler et al. [26].	2007	NR	NR	NR	NR	NR	NR	NR
Gratz et al. a [11]	2014	NR	NR	NR	NR	NR	NR	NR
Gratz et al. b [11]	2014	NR	NR	NR	NR	NR	NR	NR
Gratz et al. c [11]	2014	NR	NR	NR	NR	NR	NR	NR
Gratz et al. overall [11]	2014	NR	NR	NR	NR	NR	NR	NR
Kakuda et al. [27]	2005	NR	2 (18%)	8 (73%)	NR	NR	6 (55%) ^b^	4 (36%)
Kidwell et al. [28]	2002	NR	NR	NR	NR	NR	NR	NR
Kim et al. [29]	2006	NR	NR	NR	NR	NR	NR	NR
Kimura et al. [30]	2013	NR	NR	NR	NR	NR	NR	NR
Lee et al. [38]	2022	42 (46%)	27 (30%)	71 (78%)	NR	24 (26%) ^a^	19 (21%) ^d^	27 (30%)
Moriya et al. [31]	2012	NR	NR	NR	NR	NR	NR	NR
Nighoghossian et al. [32]	2002	NR	5 (25%)	16 (80%)	NR	7 (35%) ^a^	5 (25%) ^d^	6 (30%)
Pratz-Sanchez et al. [33]	2016	NR	NR	NR	NR	NR	NR	NR
Schlemm et al. [15]	2022	16 (16%)	38 (39%)	64 (65%)	NR	14 (14%) ^a^	NR	22 (22%)
Shi et al. [39]	2016	16 (43%)	10 (27%)	26 (70%)	11 (30%)	29 (78%)	NR	13 (35%)
Soo et al. [40]	2012	NR	20 (87%)	20 (87%)	6 (26%)	12 (52%)	12 (52%) ^d^	8 (35%)
Turc et al. [34]	2015	NR	NR	NR	NR	NR	NR	NR
Yan et al. [35]	2015	NR	NR	NR	NR	NR	NR	NR
Zand et al. [36]	2017	11 (11%)	41 (40%)	91 (88%)	NR	36 (35%) ^a^	36 (35%) ^c^	39 (38%)

Note: a, b and c correspond to cohorts receiving IVT, EVT and bridging, respectively, in the said study. Choi et al. [21] and Choi et al. (2) [22] refer to studies from 2019 and 2020, respectively. Abbreviations: CMB = cerebral microbleed, NR = not reported, *n* = number, AF = atrial fibrillation, HL = hyperlipidaemia, HTN = hypertension, CAD = coronary artery disease, PS = prior stroke, TIA = transient ischaemic attack, DM = diabetes mellitus, ^a^: Prior stroke only, ^b^: current and previous smoking, ^c^: current smoking, ^d^: previous/current smoking not specified.

**Table 3 biomedicines-11-02865-t003:** Clinical outcomes of studies selected for meta-analysis.

Author	Reperfusion Therapy	sICH (*n*, %)			HT			mRS 3–6 at 90 Days (*n*, %)			Mortality at 90 Days (*n*, %)		
		Overall	CMB	No-CMB	Overall	CMB	No-CMB	Overall	CMB	No-CMB	Overall	CMB	No-CMB
Bai et al. [17]	IVT	NR	NR	NR	13 (11.5)	NR	NR	NR	NR	NR	NR	NR	NR
Brauner et al. [37]	EVT	22 (9.1)	NR	NR	NR	NR	NR	106 (49.5)	NR	NR	NR	NR	NR
Capuana et al. a [18] (SITS-MOST)	IVT	2 (0.46)	2 (0.46)	0 (0)	NR	NR	NR	130 (33.5)	NR	NR	33 (8.5)	NR	NR
Capuana et al. b [18] (ECASS)	IVT	28 (6.45)	11 (2.53)	17 (3.92)	NR	NR	NR	130 (33.5)	NR	NR	33 (8.5)	NR	NR
Capuana et al. c [18] (NINDS)	IVT	13 (3.00)	7 (1.61)	6 (1.38)	NR	NR	NR	130 (33.5)	NR	NR	33 (8.5)	NR	NR
Capuana et al. Overall [18]	IVT	NR	NR	NR	NR	NR	NR	130 (33.5)	39 (8.99)	91 (20.97)	33 (8.5)	11 (2.53)	22 (5.07)
Chacon-Portillo et al. [19].	IVT	6 (2.05)	3 (1.03)	3 (1.03)	46 (15.75)	12 (4.11)	34 (11.64)	63 (28.8)	16 (5.48)	47 (21.5)	NR	NR	NR
Chatzikonstaninou et al. [20].	IVT	NR	NR	NR	18 (13.64)	0 (0)	18 (13.64)	NR	NR	NR	NR	NR	NR
Choi et al. a [21]	IVT	NR	NR	NR		NR	NR	NR	NR	NR	NR	NR	NR
Choi et al. b [21]	EVT	NR	NR	NR	NR	NR	NR	NR	NR	NR	NR	NR	NR
Choi et al. Overall [21]	IVT/EVT	69 (5.16)	17 (10.30)	52 (3.80)	420 (27.42)	66 (4.31)	354 (23.11)	865 (56.46)	103 (62.42)	762 (55.74)	187 (12.21)	26 (15.76)	161 (11.78)
Dannenburg et al. [23].	IVT	10 (3.07)	7 (2.15)	3 (0.92)	NR	NR	NR	NR	NR	NR	NR	NR	NR
Derex et al. [24].	IVT	3 (6.82)	1 (2.27)	2 (4.55) NR	NR	NR	NR	NR	NR	NR NR	NR	NR	NR NR
Derraz et al. [25].	EVT	66 (12.87)	15 (2.92)	51 (9.94)	NR	NR	NR	281 (54.78)	59 (11.50)	222 (43.27)	88 (17.15)	24 (4.68)	64 (12.48)
Fiehler et al. [26].	IVT	18 (3.16)	5 (0.88)	13 (2.28)	NR	NR	NR	NR	NR	NR	NR	NR	NR
Gratz et al. a [11]	IVT	6 (1.53)	0 (0)	6 (1.53)	NR	NR	NR	70 (17.86)	22 (5.61)	48 (12.24)	38 (9.69)	10 (2.55)	28 (7.14)
Gratz et al. b [11]	EVT	9 (2.30)	1 (0.26)	8 (2.04)	NR	NR	NR	93 (23.72)	26 (6.63)	67 (17.09)	47 (12.00)	16 (4.08)	31 (7.91)
Gratz et al. c [11]	Bridging	6 (1.53)	0 (0)	6 (1.53)	NR	NR	NR	30 (7.65)	4 (1.02)	26 (6.63)	15 (3.83)	1 (0.26)	14 (3.57)
Gratz et al. Overall [11]	IVT and/or EVT	21 (5.36)	1 (0.26)	20 (5.10)	NR	NR	NR	193 (49.23)	52 (13.27)	141 (35.97)	100 (25.51)	27 (6.89)	73 (18.62)
Kakuda et al. [27]	IVT	7 (9.72)	0 (0)	7 (9.72)	20 (27.78)	3 (4.29)	17 (24.29)	NR	NR	NR	NR	NR	NR
Kidwell et al. [28]	IVT	NR	NR	NR	15 (36.59)	2 (4.88)	13 (31.71)	NR	NR	NR	NR	NR	NR
Kim et al. [29]	IVT	5 (7.69)	3 (4.62)	2 (3.08)	17 (26.15)	8 (12.31)	9 (13.85)	NR	NR	NR	NR	NR	NR
Kimura et al. [30].	IVT	6 (2.46)	NR	NR	65 ()29.02	NR	NR	NR	NR	NR	NR	NR	NR
Lee et al. [38]	EVT	NR	NR	NR	170 (29.46)	32 (5.55)	138 (23.91)	288 (49.91)	59 (10.23)	229 (39.69)	NR	NR	NR
Moriya et al. [31]	IVT	NR	NR	NR	26 (36.62)	6 (8.54)	20 (28.17)	NR	NR	NR	NR	NR	NR
Nighoghossian et al. [32]	IVT	NR	NR	NR	34 (34.00)	NR	NR	NR	NR	NR	NR	NR	NR
Pratz-Sanchez et al. [33]	IVT	NR	NR	NR	78 (7.86)	NR	NR	432 (43.55)	NR	NR	134 (13.51)	NR	NR
Schlemm et al. [15] a. (SITS-MOST)	IVT	6 (1.31)	2 (0.44)	4 (0.87)	102 (22.22)	30 (6.54)	72 (15.69)	128 (28.51)	37 (8.06)	91 (19.83)	10 (2.18)	4 (0.87)	6 (1.31)
Schlemm et al. [15] b. (ECASS II)	IVT	8 (1.74)	3 (0.65)	5 (1.09)	102 (22.22)	30 (6.54)	72 (15.69)	128 (28.51)	37 (8.06)	91 (19.83)	10 (2.18)	4 (0.87)	6 (1.31)
Schlemm et al. [15] c. (ECASS II!)	IVT	6 (1.31)	2 (0.44)	4 (0.87)	102 (22.22)	30 (6.54)	72 (15.69)	128 (28.51)	37 (8.06)	91 (19.83)	10 (2.18)	4 (0.87)	6 (1.31)
Schlemm et al. [15] d. (NINDS)	IVT	26 (5.66)	11 (2.40)	15 (3.27)	102 (22.22)	30 (6.54)	72 (15.69)	128 (28.51)	37 (8.06)	91 (19.83)	10 (2.18)	4 (0.87)	6 (1.31)
Shi et al. [39]. *	EVT	NR	NR	NR	91 (44.14)	14 (6.80)	77 (37.38)	NR	NR	NR	NR	NR	NR
Soo et al. [40].	EVT	NR	NR	7 (5.26)	1 (0.75)	6 (4.51)	NR	NR	NR	NR	NR	NR	NR
Turc et al. [34] a. (NINDS)	IVT	65 (9.07)	NR	NR	NR	NR	NR	329 (45.89)	NR	NR	NR	NR	NR
Turc et al. [34] b. (ECASS II)	IVT	64 (8.93)	NR	NR	NR	NR	NR	329 (45.89)	NR	NR	NR	NR	NR
Turc et al. [34] c. (ECASS III)	IVT	27 (3.77)	NR	NR	NR	NR	NR	329 (45.89)	NR	NR	NR	NR	NR
Turc et al. [34] d. (SITS-MOST)	IVT	27 (3.77)	NR	NR	NR	NR	NR	329 (45.89)	NR	NR	NR	NR	NR
Yan et al. [35]	IVT	2 (1.65)	NR	NR	36 (29.75)	NR	NR	NR	NR	NR	NR	NR	NR
Zand et al. [36] *	IVT	25 (3.72)	5 (0.74)	20 (2.98)	NR	NR	NR	NR	NR	NR	NR	NR	NR

Note: * Shi et al. [39] and Zand et al. [36] reported in hospital mortality, not 90-day mortality. Capuana et al. [18] a, b, c corresponds to cohorts assessed for sICH using the SITS-MOST, ECASS-II and NINDS criteria, respectively, in the said study. Schlemm et al. [15] a, b, c, d correspond to cohorts assessed for sICH using the SITS-MOST, ECASS-II, ECASS-III, and NINDS, respectively. Choi et al. [21] a, b and overall correspond to cohorts receiving IVT, EVT and IVT and/or EVT, respectively, within this study. Gratz et al. [11] a, b, c and overall represents cohorts receiving IVT, EVT, Bridging and IVT and/or EVT, respectively, within this study. Turc et al. [34] a, b, c and d correspond to cohorts assessed for sICH using the NINDS, ECASS-II, ECASS-III and SITS-MOST criteria, respectively, in the said study. Abbreviations: CMB = cerebral microbleed, sICH = symptomatic intracerebral haemorrhage, HT = haemorrhagic transformation, IVT = intravenous thrombolysis, EVT = endovascular thrombectomy, NR = not reported, NIHSS = National Institutes of Health Stroke Scale, ECASS = European Cooperative Acute Stroke Study, ECASS-II = second European Cooperative Acute Stroke Study, ECASS-III = third European Cooperative Acute Stroke Study, NINDS = National Institute of Neurological Disorders and Stroke, SITS-MOST = Safe Implementation of Thrombolysis in Stroke-Monitoring Study.

**Table 4 biomedicines-11-02865-t004:** Summary effects and heterogeneity obtained from the meta-analysis of the association of cerebral microbleeds with clinical outcomes in acute ischaemic stroke patients who underwent reperfusion therapy.

Outcome	Reperfusion Therapy	Effect Measure	Test of ES = 0	Summary Effects	Heterogeneity ^⍺^			Heterogeneity Variance Estimates			
				REDL	Tests of Overall Effect	Cochran’s Q	Chi-Squared	H	I^2^ ≤ *	*p*-Value	τ^2^ ≤ ^†^
				OR (95% CI)							
sICH	IVT	OR	N/A	2.57 [1.82; 3.61]	*p* < 0.0001 z = 5.416	9.37	N/A	N/A	0.0%	0.806	N/A
	EVT	OR	N/A	1.14 [0.40; 3.21]	*p* = 0.805 z = 0.246	1.39	N/A	N/A	28.1%	0.238	N/A
	IVT/EVT	OR	N/A	8.96 [4.82; 16.63]	*p* < 0.0001 z = 6.945	0.00	N/A	N/A	NR	NR	N/A
	Bridging	OR	N/A	0.67 [0.03; 13.28]	*p* = 0.792 z = −0.264	0.00	N/A	N/A	NR	NR	N/A
	Overall	OR	N/A	2.57 [1.72; 3.83]	*p* < 0.001 z = 4.634	31.33	N/A	1.319 [1.000; 1.782]	68.5%	0.026	0.2762
HT	IVT	OR	N/A	1.46 [1.03; 2.07]	*p* = 0.034 z = 2.125	2.17	N/A	N/A	0.0%	0.903	N/A
	EVT	OR	N/A	1.19 [0.81; 1.76]	*p* = 0.373 z = 0.373	1.00	N/A	N/A	0.0%	0.606	N/A
	IVT/EVT	OR	N/A	1.92 [1.37; 2.68]	*p* < 0.0001 z = 3.811	0.00	N/A	N/A	NR	NR	N/A
	Bridging	OR	N/A	N/A	N/A	N/A	N/A	N/A	N/A	N/A	N/A
	Overall	OR	N/A	1.53 [1.25; 1.88]	*p* < 0.0001 z = 4.097	6.54	N/A	0.828 [1.000; 1.212]	0.0%	0.768	<0.0001
Poor Functional Outcome at 90 Days	IVT	OR	N/A	1.70 [1.28; 2.25]	*p* < 0.0001 z = 3.709	2.23	N/A	N/A	0.0%	0.526	N/A
	EVT	OR	N/A	1.70 [1.26; 2.29]	*p* = 0.001 z = 3.453	1.98	N/A	N/A	0.0%	0.372	N/A
	IVT/EVT	OR	N/A	1.32 [0.95; 1.84]	*p* = 0.103 z = 1.631	0.00	N/A	N/A	NR	NR	N/A
	Bridging	OR	N/A	2.77 [0.47; 16.27]	*p* = 0.260 z = 1.127	0.00	N/A	N/A	NR	NR	N/A
	Overall	OR	N/A	1.59 [1.34; 1.89]	*p* < 0.001 z = 5.257	6.19	N/A	0.880 [1.000; 1.309]	41.7%	0.626	<0.0001
Mortality at 90 Days	IVT	OR	N/A	1.52 [0.91; 2.54]	*p* = 0.109 z = 1.603	1.11	N/A	N/A	0.0%	0.573	N/A
	EVT	OR	N/A	2.14 [1.37; 3.34]	*p* = 0.001 z = 3.364	0.04	N/A	N/A	0.0%	0.846	N/A
	IVT/EVT	OR	N/A	1.40 [0.89; 2.20]	*p* = 0.142 z = 1.469	0.00	N/A	N/A	NR	NR	N/A
	Bridging	OR	N/A	0.69 [0.07; 6.37]	*p* = 0.740 z = −0.332	0.00	N/A	N/A	NR	NR	N/A
	Overall	OR	N/A	1.65 [1.27; 2.16]	*p* < 0.001 z = 3.692	3.67	N/A	0.783 [1.000; 1.255]	0.0%	0.721	<0.0001
Prevalence	IVT	Prevalence	*p* < 0.001 z = 14.62	0.20 [0.16; 0.25]	N/A	N/A	417.94	N/A	95.21%	<0.001	0.06
	EVT	Prevalence	*p* < 0.001 z = 12.98	0.18 [0.14; 0.23]	N/A	N/A	75.85	N/A	90.77%	<0.001	0.03
	Bridging	Prevalence	*p* < 0.001 z = 4.16	N/A	N/A	N/A	N/A	N/A	N/A	N/A	N/A
	Overall	Prevalence	*p* < 0.001 z = 19.06	0.19 [0.16; 0.23]	N/A	N/A	501.52	N/A	94.02%	<0.001	0.03

Note: In IVT/EVT Reperfusion Subgroup, the study patients received reperfusion therapy, however, it didn’t specify how many patients received specific reperfusion treatment: IVT or EVT. Abbreviations: sICH = symptomatic intracerebral haemorrhage, HT = haemorrhagic transformation, IVT = intravenous thrombolysis, EVT = endovascular thrombectomy, OR = odds ratio, CI = confidence interval, NR = not reported, N/A = not applicable, REDL = DerSimonian and Laird Random Effects method, Q = heterogeneity measures were calculated from data with 95% confidence intervals (95% CIs), based on non-central χ2 (common effect) distribution for Cochran’s Q test, H = relative excess in Cochran’s Q divided by degrees of freedom, I^2^ = proportion of total variation in effect estimate between study heterogeneity (based on Cochran’s Q test), τ^2^ = between-study variance to test the heterogeneity among subgroups, * = numbers in I^2^ ≤ are percentages; ^⍺^ = heterogeneity measures were calculated from the data with 95% CIs based on gamma (random effects) distribution for Q, ^†^—heterogeneity variance estimates (tau≤) were derived from the DerSimonian and Laird method.

## Data Availability

The original contributions presented in the study are included in the article and online Appendix A, and further inquiries can be directed to the corresponding author.

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
