# Peer review of "Prevalence and Impact of Cerebral Microbleeds on Clinical and Safety Outcomes in Acute Ischaemic Stroke Patients Receiving Reperfusion Therapy: A Systematic Review and Meta-Analysis"

_biomedicines, 2023, doi:10.3390/biomedicines11102865_

Round 1

Reviewer 1 Report

The aim of the study was to conduct an updated systematic review and meta-analysis of observational studies to investigate the relationships and impact of cerebral microbleeds (CMBs) on clinical and safety outcomes in acute ischemic stroke patients receiving reperfusion therapy. The authors found  a significant association between CMBs and adverse postprocedural safety outcomes encompassing symptomatic intracerebral hemorrhage (sICH), poor functional outcome, and increased mortality, although not with hemorrhagic transformation (HT). The study is potentially  interesting, but can be improved if the following considerations are addressed:    

1.       The authors should clearly mention in the Introduction that signs of cerebral small vessel disease on conventional MRI include: cerebral microbleeds, leukoaraiosis, recent subcortical lacunar infarcts (clinically symptomatic), lacunes (clinically silent), prominent perivascular spaces, and atrophy (Int J Mol Sci 2022; 23, 1497).  

2.       Did the authors find clinical differences with respect to gender? The impact of female gender can be observed in the distribution of risk factors, stroke subtype, stroke severity, and outcome (Clin Neurol Neurosurg 2014 Dec;127:19-24). Did the authors take this into account in their study protocol?.   

3.         It would be interesting for the authors to clearly emphasize the need for future clinical studies on the prevalence and impact of cerebral microbleeds on clinical and safety outcomes in acute ischemic stroke patients receiving reperfusion therapy in the subgroup of very old patients ( aged 80 years or older) since demographics and risk factors are quite different in this age segment of stroke patients (see Clin Neurol Neurosurg 2006; 108: 638-643). 

Author Response

We would like to express our appreciation for your thoughtful review of our study, titled " Prevalence and Impact of Cerebral Microbleeds on Clinical and Safety Outcomes in Acute Ischaemic Stroke Patients Receiving Reperfusion Therapy: A Systematic Review and Meta-Analysis." Your constructive feedback has been invaluable in improving our manuscript. Our point-by-point rebuttal to individual points are provided below:

Q1: The authors should clearly mention in the Introduction that signs of cerebral small vessel disease on conventional MRI include: cerebral microbleeds, leukoaraiosis, recent subcortical lacunar infarcts (clinically symptomatic), lacunes (clinically silent), prominent perivascular spaces, and atrophy (Int J Mol Sci 2022; 23, 1497).

Reply: We acknowledge this point regarding the signs of cerebral small vessel disease on conventional MRI. As suggested, we have included the following information in our Introduction. The reference has also been cited.

Page 2 [Lines 52-55]

Conventional magnetic resonance imaging (MRI) plays a crucial role in the detection and evaluation of cerebral small vessel disease (CSVD). Key MRI findings indicative of CSVD encompass CMBs, leukoaraiosis, recent clinically symptomatic subcortical lacunar infarcts, clinically silent lacunes, conspicuous perivascular spaces, and atrophy[4].

Q2: Did the authors find clinical differences with respect to gender? The impact of female gender can be observed in the distribution of risk factors, stroke subtype, stroke severity, and outcome (Clin Neurol Neurosurg 2014 Dec;127:19-24). Did the authors take this into account in their study protocol?.

Reply: Thank you for raising this pertinent question regarding the potential influence of gender on clinical outcomes. We agree that gender-related differences can significantly impact the distribution of risk factors, stroke subtype, stroke severity, and ultimately, patient outcomes. In our study, we did not specifically address this aspect. However, as suggested, we have included the following statement to our Limitations.

Page 8 [Page 478-80]

Gender disparities may exist in the distribution of risk factors, stroke subtype, stroke severity, and post-stroke outcomes [95]. Nonetheless, this study did not explore this and merits further investigation.

Q3: It would be interesting for the authors to clearly emphasize the need for future clinical studies on the prevalence and impact of cerebral microbleeds on clinical and safety outcomes in acute ischemic stroke patients receiving reperfusion therapy in the subgroup of very old patients ( aged 80 years or older) since demographics and risk factors are quite different in this age segment of stroke patients (see Clin Neurol Neurosurg 2006; 108: 638-643).

Reply: We note your suggestion to highlight the need for future clinical studies on cerebral microbleeds in very old patients (aged 80 years or older). Demographics and risk factors indeed differ in this age segment of stroke patients, and exploring the prevalence and impact of cerebral microbleeds in this population is a valuable area for further research. We have included a statement in our discussion section emphasizing the importance of future studies in this specific subgroup.

Pages 7-8 [Lines 446-456]

Given the distinct demographics and risk factors in very old age segment (aged 80 years or older) of stroke patients, future clinical studies are warranted to investigate the prevalence and effects of CMBs on clinical and safety outcomes in the specific subgroup of AIS patients [89].

Once again, we sincerely appreciate your feedback, and we hope the changes made in our revised manuscript reflect these considerations.

Reviewer 2 Report

This is an excellent meta-analysis. I have read the paper with a great interest. As a reviewer I have to give some suggestions. 

Minor

Please, try to arrange Figure 2 in this way that groups numbers (N, C) did not come one on the other. You may get some extra space  by moving p-value on the right side of the Fig. after ES and 95% CIs, for instance. Then, You will have some more space to separate collumns with groups' numbers.

It would be great if the Authors could show (in Discussion section) just published study on CMBs in patients with AIS related to atrial fibrillation and the outcomes with oral anticoagulation ( https://doi.org/10.3390/jcm12175752 )

Author Response

Thank you for your positive feedback on our meta-analysis, and we appreciate your valuable suggestions. We have made best of our efforts to incorporate the comments.

Q1: Please, try to arrange Figure 2 in this way that groups numbers (N, C) did not come one on the other. You may get some extra space by moving p-value on the right side of the Fig. after ES and 95% CIs, for instance. Then, You will have some more space to separate collumns with groups' numbers.

Reply: We thank the reviewer for the comment. We have adjusted the layout as per the suggestion. We hope the revised Figure (which has now been inserted replacing the old one) will make the presentation more legible and prevent overlap.

Q2: It would be great if the Authors could show (in Discussion section) just published study on CMBs in patients with AIS related to atrial fibrillation and the outcomes with oral anticoagulation ( https://doi.org/10.3390/jcm12175752 ).

Reply: We appreciate your suggestion to include an update based on recently published study. We have incorporated the following paragraph within the context of our meta-analysis.

Page 8 [Lines 50-458]

Resuming anticoagulation, and its optimal timing, after an ICH in patients with atrial fibrillation (AF) poses an ongoing clinical conundrum[90]. Individualised treatment decision making that factors in potential for future bleeding events and the risk of thromboembolic complications is suggested that takes into account various risk factors, including blood pressure control, age, the location of the ICH, the presence of CSVD markers (such as CMBs, leukoaraiosis, and cortical superficial siderosis), and the indication for antiplatelet therapy [91].

Finally, once again, thank you for the feedback. It is invaluable in improving the comprehensiveness of our work, and we are grateful for your time and effort in reviewing our manuscript. We have also made some minor changes to improve the formatting and presentation of our Tables.
